# Quantization Variation: A New Perspective on Training Transformers with Low-Bit Precision

**Xijie Huang**[1], **Zhiqiang Shen**[2], **Pingcheng Dong**[1], **Tim Kwang-Ting CHENG**[1]

[1]*Hong Kong University of Science and Technology,* [2]*Mohamed bin Zayed University of Artificial Intelligence*
*xhuangbs@connect.ust.hk, Zhiqiang.Shen@mbzuai.ac.ae, pingcheng.dong@connect.ust.hk, timcheng@ust.hk*

**Reviewed on OpenReview:** *https://openreview.net/forum?id=MHfoAOQf6g*

## Abstract

Despite the outstanding performance of transformers in both language and vision tasks, the expanding computation and model size have increased the demand for efficient deployment. To address the heavy computation and parameter drawbacks, quantization is frequently studied in the community as a representative model compression technique and has seen extensive use on ConvNets. However, due to the unique properties of transformers, the extreme low-bit quantization applications are still limited and underexplored. In this paper, we identify the difficulty of transformer-based low-bit quantization-aware training on its unique **variation** behaviors, which significantly differ from ConvNets. The term **variation** is defined based on comprehensive quantitative analysis in three hierarchies: various module quantization sensitivities, outliers in static weight and activation distribution, and oscillation in dynamic parameter fluctuations. These variations of transformers bring instability to the quantization-aware training (QAT) and negatively influence the performance. We explore the best practices to alleviate the variation's influence during low-bit transformer QAT and propose a variation-aware quantization scheme for both vision and language transformers. We extensively verify and demonstrate our scheme can alleviate the variation and improve the performance of transformers across various models and tasks. For the 2-bit Swin-T and binary BERT-base, our solutions achieve a **3.35%** and **1.4%** accuracy improvement over previous state-of-the-art methods on the ImageNet-1K dataset and GLUE benchmark. Codes and models are available at https://github.com/HuangOwen/Quantization-Variation.

## 1 Introduction

Transformer-based models have achieved impressive accuracy across multiple modalities including a variety of computer vision (Zhang et al., 2020a;b; Kirillov et al., 2023), natural language (Devlin et al., 2018; Chowdhery et al., 2022; Touvron et al., 2023; Achiam et al., 2023), acoustic and speech Di et al. (2021); Li et al. (2023a); Gao et al. (2024) tasks. Despite the intrinsic superiority of transformer architecture, their remarkable performance also comes from the parameter numbers. For instance, Swin-L (Liu et al., 2021a) has a total number of parameters of 197M with FLOPs of 34.5G. Dehghani et al. (2023) scales vision transformers to 22B for better performance. The language model GPT-3 (Brown et al., 2020)

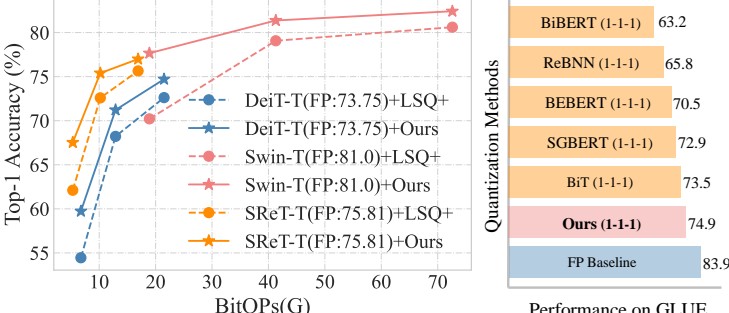

Figure 1: **Left:** ImageNet-1K Top-1 accuracy vs. BitOPs comparison of 2/3/4-bit quantized ViT models using LSQ+ (Bhalgat et al., 2020) and our method. **Right:** GLUE performance comparison of different binary (1-1-1-bit) BERT models.

boasts 175 billion parameters. As a result of the tremendous parameter numbers, high latency and large model sizes have become the most significant obstacles to the efficient deployment of transformers, especially on devices with computation constraints.

In recent years, researchers have explored and proposed various model compression methods to improve the computational efficiency of deep learning models. These model compression techniques include quantization (Bhalgat et al., 2020; Huang et al., 2022; Ding et al., 2022; Tang et al., 2023; Xiao et al., 2023a), pruning (Liu et al., 2017; 2018; 2019b; Kim et al., 2023; Ma et al., 2023), knowledge distillation (Hinton et al., 2015; Shen & Xing, 2021), and compact network design (Howard et al., 2017; Pham et al., 2018; Guo et al., 2020). Among these methods, quantization of weights and activations have been the most widely utilized techniques because they enjoy the advantage of the promising affinity across different hardware architectures (Judd et al., 2016; Jouppi et al., 2017; Sharma et al., 2018). Although efforts (Liu et al., 2021c; Yuan et al., 2021; Lin et al., 2021; Li et al., 2022c; Ding et al., 2022; Liu et al., 2023b; Frantar et al., 2023b; Wei et al., 2023) have been made to apply quantization techniques to transformers, most of them are based on Post-Training Quantization (PTQ) which suffers from a significant decline in performance and a bitwidth limitation at 8-bit or 6-bit. Additionally, the few existing Quantization-Aware Training (QAT) methods (Li et al., 2022b; Li & Gu, 2022; Li et al., 2022a; Tao et al., 2022; Liu et al., 2023c; Yu et al., 2023; Dong et al., 2023) take significantly more time than the full-precision model in training, and the models still fail to achieve the desired performance when being quantized to extreme low-bit precision such as 2-bit and binary.

The higher degradation in accuracy of quantized transformers compared to ConvNets guides us to raise the question: *What is it that hinders us from improving the performance of low-bit quantized transformers?* Meanwhile, the low efficiency of previous QAT methods makes applying quantization to more transformer structures difficult. Thus, another question we would like to raise is: *How to improve the efficiency of transformer quantization?*

To comprehensively decipher the inherent obstacles that adversely impact the efficacy and performance of transformer quantization, in this work, we research the unique **variation** behavior of transformers in different hierarchies. The terminology **variation** in our paper refers to both the various quantization sensitivity of different modules (described in Section 4.1), outliers in weight/activation distribution (as described in Section 4.2), and weight updating instability during QAT (as described in Section 4.3). We empirically find that these challenges are inherently inter-connected, thus we use term **variation** to summarize them. We initially conducted an exhaustive investigation of the quantization resilience of each component within the structural layout of the transformer. The empirical findings derived from the quantization ablation experiments show that specific components, such as Multi-head self-attention (MHSA), exhibit higher sensitivity to quantization than others. We further compare the weight and activation distribution between transformers and ConvNets, deducing that the distribution outliers serve as the pivotal factor instigating complications with respect to transformer quantization. Through constant monitoring of the weight-changing trajectory during the training phase, we revealed that variation in distribution instigates the variation in weight updates known as weight oscillation. Such a phenomenon has detrimental effects on quantization, potentially culminating in decelerated convergence. Different from previous research Xiao et al. (2023a); Wei et al. (2022); Lin et al. (2023) to analyze the outlier of distribution in PTQ, we are the first to provide novel and quantitative analysis on the variations in transformer QAT and reveal their inner causality.

Based on the variation analysis, we propose an optimized solution for transformer quantization that is attuned to variations, demonstrating enhanced efficiency. In terms of the sensitivity distribution variance observed across differing modules, we introduce a module-specific scaling methodology. This strategy seeks to identify varying scale factors pertinent to different modules, thereby holistically accommodating the diversity in weight distribution through a gradient scaling technique that is sensitive to weight magnitude. When compared with the baseline quantization method, LSQ+ (Bhalgat et al., 2020), the presented approach exhibits less susceptibility to fluctuations in weight distribution and outliers that may arise within transformers. For Vision Transformers (ViTs), a multi-crop knowledge distillation approach is employed, which aids in decreasing the data variance within mini-batches during the training phase, thereby stabilizing and expediting the training process. Furthermore, to combat the potential oscillation throughout the training phase, we put forth a regularizer that is attuned to oscillation within quantization bins. This process seeks to penalize the variance in weight distribution within each respective quantization bin.

Extensive experiments across various transformer architectures with different characteristics, including DeiT (Touvron et al., 2021), Swin Transformer (Liu et al., 2021a), SReT (Shen et al., 2021a), and BERT (Devlin et al., 2018), are conducted to verify the effectiveness and efficiency of our proposed method. As shown in Figure 1, for DeiT-T on ImageNet-1K dataset, our 4-bit quantized model can significantly improve top-1 accuracy to 74.71% compared to the model quantized by LSQ+ (Bhalgat et al., 2020) which achieves 72.62%. Furthermore, our binarized BERT-based model achieves 74.9% average accuracy on the GLUE benchmark, surpassing the previous state-of-the-art method by 1.4%. Through these methodologies, we exhibit exceptional training optimization, as evidenced by a 50% reduction in total training time compared to our established baseline. In summary, our contribution can be concluded as:

- We are the first to reveal the inherent complexity associated with low-bit quantization of transformers from the perspective of **variation**. Our claims that **variations** lurk in multiple hierarchies of transformers are substantiated through sensitivity analysis, a comparison of transformers to ConvNets, and an investigation of oscillatory behavior.

- We explore the best practices to reduce the **variation** in transformer quantization, including module-dependent quantization, oscillation-aware regularization, and a novel multi-crop knowledge distillation scheme designed for ViTs.

- We perform extensive experiments of DeiT, Swin, SReT, and BERT across multiple modalities, including the ImageNet-1K dataset and GLUE benchmark. Our approach significantly reduces **variation** and results in both superior efficiency and performance over prior state-of-the-art schemes.

## 2 Related Work

**Transformer-based Models:** Transformer (Vaswani et al., 2017) was proposed based on an attention mechanism and demonstrated remarkable performance across various benchmarks such as GLUE (Wang et al., 2018) and SQuAD (Rajpurkar et al., 2016). BERT (Devlin et al., 2018), RoBERTa (Liu et al., 2019a), XL-Net Yang et al. (2019), GPT Achiam et al. (2023), LLaMA Touvron et al. (2023) have served as an essential building block in modern NLP pipelines. Inspired by the success in NLP, Vision Transformers(ViTs) (Dosovitskiy et al., 2020) utilize multi-head self-attention treating an image as patches/tokens. The attention mechanism can help capture both short-range and long-range visual dependencies. Various extensions of ViTs (Touvron et al., 2021; Liu et al., 2021a; Wu et al., 2021; Dong et al., 2022; Hatamizadeh et al., 2023) and more applications (Zheng et al., 2021; Arnab et al., 2021) are still emerging.

**Quantization Techniques:** Quantization techniques aim to replace the full-precision weights and activations with lower-precision representation. Based on the quantization intervals, they can be categorized into uniform and non-uniform quantization. While uniform quantization (Huang et al., 2022) with uniform quantization interval has better hardware affinity and efficiency, Non-uniform quantization (Li et al., 2019), due to its flexible representation, can usually better allocate the quantization values to minimize the quantization error and achieve better performance. In addition, the quantization methods can also be classified as quantization-aware training (QAT) (Bhalgat et al., 2020; Liu et al., 2023a;c) and post-training quantization (PTQ) (Nagel et al., 2020; Fang et al., 2020; Wang et al., 2020; Lee et al., 2023; Xiao et al., 2023a) based on whether to retrain a model with quantized weights and activations or start with a pre-trained model and directly quantize it without extra training. The majority of previous transformer quantization methods, such as Liu et al. (2021c), PTQ4ViT (Yuan et al., 2021), FQ-ViT (Lin et al., 2021), ZeroQuant (Yao et al., 2022), and NoisyQuant (Liu et al., 2023b) focused on PTQ of transformers. Due to the intrinsic restriction of PTQ, these methods only perform 8-bit or 6-bit quantization. In this work, we focus on low-bit ($\leq$ 4-bit) uniform QAT setting.

**Knowledge Distillation:** The concept of knowledge distillation is first proposed in (Hinton et al., 2015), where the core insight is to encourage student models to emulate the distribution of teacher models' prediction. The prediction distribution of teacher models contains more information than the one-hot labels. More recently, various knowledge distillation methods (Cho & Hariharan, 2019; Park et al., 2019; Tung & Mori, 2019; Mirzadeh et al., 2020; Shen & Xing, 2021; Li et al., 2023b) have been proposed for better efficiency

and effectiveness. The knowledge-distillation methods are also widely adopted in previous research (Mishra & Marr, 2018; Polino et al., 2018; Huang et al., 2022; Liu et al., 2023a) to help quantization-aware training.

## 3 Preliminaries

**Transformer Architecture.** The basic block of the transformers is the transformer layer, consisting of Multi-head Self Attention (MHSA), Layer Normalization (LN) (Ba et al., 2016), and Feed-forward Network (FFN). The transformer layer can be formulated as

$$\mathbf{X}' = \text{LN}(\mathbf{X_i} + \text{MHSA}(\mathbf{X_i})), \quad \mathbf{X_O} = \text{LN}(\mathbf{X}' + \text{FFN}(\mathbf{X}')), \tag{1}$$

where $\mathbf{X_i}$, $\mathbf{X}'$, and $\mathbf{X_o}$ are this transformer block's input, intermediate representation, and output. The MHSA module consists of $h$ heads, and each head performs inner products with a scaling factor and a *softmax* operation. For the $i$-th head, input $\mathbf{X_i}$ is projected into *query*, *key*, and *value* vectors with multiplication with learnable weight matrix $\mathbf{W_{Q,i}}, \mathbf{W_{K,i}}, \mathbf{W_{V,i}}$, which can be written as:

$$\mathbf{Q_i} = \mathbf{X_i}\mathbf{W_{Q,i}}, \quad \mathbf{K_i} = \mathbf{X_i}\mathbf{W_{K,i}}, \quad \mathbf{V_i} = \mathbf{X_i}\mathbf{W_{V,i}}, \tag{2}$$

and the output of $i$-th head is

$$\text{head}_\mathbf{i} = \text{softmax}(\mathbf{Q_i}\mathbf{K_i^T}/\sqrt{\mathbf{d_k}})\mathbf{V_i}, \tag{3}$$

where $1/\sqrt{\mathbf{d_k}}$ is the scaling factor for normalization. MHSA further concatenates the output of these heads to improve the representative capacity and projects to the output by multiplication with a learnable weight matrix $\mathbf{W_o}$:

$$\text{MHSA}(\mathbf{X_i}) = \text{Concat}(\text{head}_\mathbf{1}, \text{head}_\mathbf{2}, ..., \text{head}_\mathbf{h})\mathbf{W_o}. \tag{4}$$

**Quantization.** Given the real-value data to be quantized as $x^r$, the scale factor $s$ of the quantizer, the number of positive quantization levels $Q_P$, and the number of negative quantization levels $Q_N$, we can have the quantizer $q_b$ that output the $b$-bit quantized representation of the input real value as $x^q = q_b(x^r)$ :

$$x^q = q_b(x^r) = s \times \lfloor \text{clip}(x^r/s, -Q_N, Q_P) \rceil, \tag{5}$$

where $\lfloor \cdot \rceil$ is the rounding function that rounds the input to the nearest integer, $\text{clip}(x, r_1, r_2)$ return $x$ with all value below $r_1$ set to be $r_1$ and all values above $r_2$ set to be $r_2$. For the unsigned quantization, $Q_N = 0, Q_P = 2^b - 1$. While for the quantization of signed data, $Q_N = 2^{b-1}, Q_P = 2^{b-1} - 1$. To solve the problem that the gradient cannot back-propagate in Equation 5, the straight-through estimator (STE) (Bengio et al., 2013) is utilized to approximate the gradient during quantization-aware training. The gradient of the rounding operation is approximated as 1 in the quantization limit. In the back-propagation with STE, the gradient of the loss $\mathcal{L}$ with respect to the real-value data $x^r$ is set to be:

$$\frac{\partial\mathcal{L}}{\partial x^r} = \frac{\partial\mathcal{L}}{\partial x^q} \cdot \mathbf{1}_{-Q_N \leq x^r/s \leq Q_P}, \tag{6}$$

where $\mathbf{1}$ is the indicator function that outputs 1 within the quantization limit and 0 otherwise. This STE is widely used in quantization-aware training and we can derive the gradient of the quantized value $x^q$ with respect to the scale factor $s$ as

$$\frac{\partial x^q}{\partial s} = \begin{cases} x^r/s + \lfloor x^r/s \rceil & \text{if } x^r/s \in (-Q_N, Q_P) \\ -Q_N & \text{if } x^r/s \in (-\infty, -Q_N] \\ Q_P & \text{if } x^r/s \in [Q_P, \infty) \end{cases} \tag{7}$$

## 4 Understanding Quantization Variation of Transformers

Many existing studies highlight that transformers exhibit greater sensitivity to quantization compared to ConvNets. For instance, Bit-Split (Wang et al., 2020), which successfully achieves 4-bit quantization on

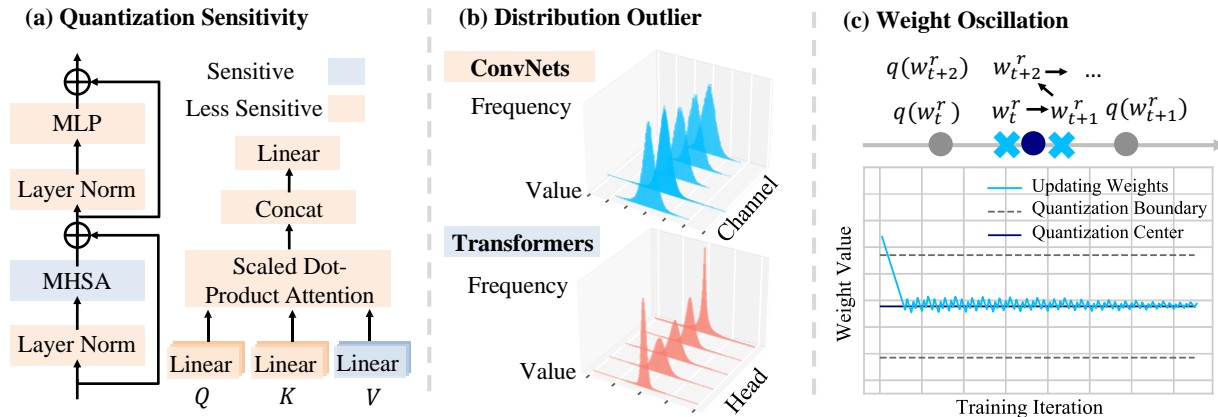

Figure 2: An overview of the variation in transformers of different hierarchies: various quantization sensitivities of different modules, outlier in weight and activation distributions, and oscillation phenomenon in dynamic parameter updates.

ResNet with an accuracy loss of less than 1%, exhibits significant accuracy degradation of over 2% (Lin et al., 2021) when applied to 8-bit quantization of DeiT on the same ImageNet-1K dataset. However, there is a paucity of analyses detailing the reasons behind transformers' heightened sensitivity compared to ConvNets. In this section, we will give a comprehensive analysis of the fundamental challenge in low-bit quantization of transformers referred to in this work as **variation**. As depicted in Figure 2, we define the term **variation** to include three components: (1) different quantization sensitivity of each module, (2) variance of weight and activation frequency distribution compared to ConvNets, and (3) abnormal weight oscillation phenomenon during QAT. We will explore the variation in sensitivity in Section 4.1 and delve into the variation in distribution and its subsequent side-effect of oscillation phenomenon in Sections 4.2 and 4.3.

## 4.1 Quantization Sensitivity

Prior study Li et al. (2022b) conducted a quantization robustness analysis on transformers, concluding that the GELU activation function substantially mitigates performance during the quantization process. However, their experiments relied on post-training quantization (PTQ), which stands in stark contrast to quantization-aware training (QAT). Moreover, their experimental methodology lacked a comprehensive analysis of different components at a more granular level, such as the quantization impact on query, key, and value weight matrices. In this section, we aim to disentangle the intricacies of transformer quantization by executing an in-depth leave-one-out analysis employing low-bit QAT.

Table 1: Leave-one-out-analysis for quantization of various components in DeiT-T on ImageNet-1K. The Para(%) stands for the percentage of parameters that are **not** quantized among all trainable parameters.

| Quantization Target | DeiT-T (W3A3) | | Swin-T (W2A2) | | SReT-T (W3A3) | |
|---|---|---|---|---|---|---|
| | Top-1 Acc(%) | Top-5 Acc(%) | Top-1 Acc(%) | Top-5 Acc(%) | Top-1 Acc(%) | Top-5 Acc(%) |
| None (FP Model) | 73.75 | 91.87 | 81.00 | 95.25 | 75.81 | 91.74 |
| All | 68.22 | 88.56 | 70.21 | 85.50 | 72.59 | 89.90 |
| All, except FFN | 69.47 | 89.60 | 72.77 | 88.02 | 73.06 | 90.22 |
| All, except MHSA | **71.28** | **90.66** | **77.93** | **92.59** | **75.10** | **91.21** |
| All, except *query* in MHSA | 69.66 | 89.94 | 73.15 | 89.74 | 73.66 | 90.43 |
| All, except *key* in MHSA | 69.92 | 89.81 | 73.09 | 89.98 | 73.58 | 90.37 |
| All, except *value* in MHSA | **70.72** | **90.40** | **75.80** | **90.66** | **74.15** | **90.84** |

In terms of quantization methods, we employ LSQ+(Bhalgat et al., 2020). All components except for the analysis target will be quantized, while the analysis target will be retained at full precision. The results of DeiT-T (3-bit), Swin-T (2-bit), and SReT-T (3-bit) on the ImageNet-1K are presented in Table 1. The results across various models and bitwidth settings indicate that MHSA, particularly the *value* weight matrices, are

highly susceptible to quantization. Although MHSA and the *value* weight matrix constitute a relatively minor fraction of parameters in comparison to the FFN, maintaining these parts at full precision can optimize the performance of the quantized model. We also provide a quantize-one-module-only analysis shown in Table 9 in the Appendix, which also comes to the same conclusion.

While we have fully exploited the clue that the quantization sensitivity of MHSA is higher than other components in transformers, another critical clue is that some heads in MHSA are more important than other heads in Transformer-based models. To empirically verify this clue, we apply an analysis to quantize various heads in different layers in transformers. The target heads are quantized to 2-bit while the remaining components are quantized to 8-bit. The results of DeiT-T with three heads in a layer and 12 layers are shown in Figure 3. The results that some heads have higher accuracy degradation show that the quantization sensitivity of different heads at different layers varies. The first and last few layers are

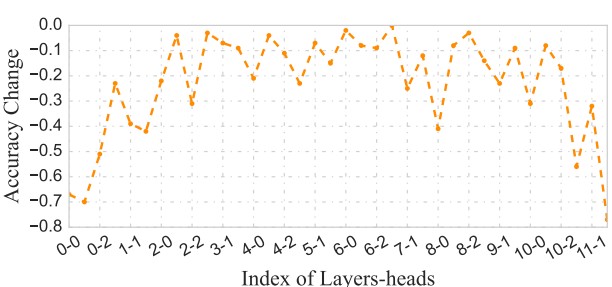

Figure 3: The accuracy degradation compared to the full-precision model when a specific head in a layer of Transformer is quantized. The label *h-l* in abscissa indicates the head *h* in layer *l* is quantized.

more sensitive to quantization. Additionally, the heads in the same layer show a variation in quantization robustness. For example, in layer 8 of the quantized model, the lower precision of head 0 (shown in 8-0 in Figure 3) will result in a higher accuracy drop compared to the two parallel heads in the same layer.

## 4.2 Distribution Outlier

In Section 4.1, we have demonstrated that transformers suffer from significant variation in the sensitivity to quantization. However, previous mixed precision quantization research on ConvNet has also discovered that different parts of models have various quantization robustness. To fully understand why the sensitivity to quantization in transformers is higher than ConvNets, we visualize and quantify the distribution of different modules inside full-precision ConvNets and transformers to compare the real distribution **variation** of transformers and ConvNets as shown in Figure 2 (b).

To give an intuitive result on the variation of ConvNets and transformers, we first visualize the weight distribution across different channels in full precision ResNet-18 (He et al., 2016) and DeiT-T. The results are shown in Figure 2. Based on our investigation, the ResNet-18 model shares a similar distribution across different channels, while the weight distribution varies significantly in different modules in the transformer-based model DeiT-T. If layer-wise quantization methods that work well for ConvNets are directly applied to transformers, the variation of distribution will result in high quantization error, which is one of the reasons that the ConvNet-based method failed on low-bit transformers.

For the activation distribution, previous research (Wei et al., 2022; Bondarenko et al., 2023) has pointed out outliers appear regularly and consistently across multiple layers in transformers. To quantify the fluctuation in the latent real-valued activation magnitude, we proceed to calculate the Average Standard Deviation of the Absolute Mean (SDAM) of the real-valued activation magnitude within each module of ConvNets and transformers. The SDAM metric has been previously employed to evaluate the stability and fairness of training in prior studies (Liu et al., 2021b). The corresponding results of the SDAM comparison are tabulated in Table 2. These numerical findings corroborate that the variability associated with transformers surpasses that of ConvNets with respect to the activation distribution.

Table 2: Standard Deviation of the Absolute Mean (SDAM) of activation in ConvNets and transformers.

| Model | ResNet-18 | VGG-11 | ViT-T | DeiT-T | Swin-T |
|---|---|---|---|---|---|
| SDAM | 5.59e-2 | 3.74e-2 | 9.65e-2 | 8.35e-2 | 9.71e-2 |

Correspondingly, prior work (Lin et al., 2021) has highlighted significant disparities in the distribution of activations in transformers as opposed to ConvNets. Although these variations may augment the representational capacity of transformers, they concurrently introduce complexities when implementing quantization. Consequently, the conception and arrangement of the quantization scheme become paramount, particularly in the generation of quantization scales and the determination of clipping factors during the process of quantization-aware training.

### 4.3 Weight Oscillation in Training

High variation in weight and activation distribution can lead to suboptimal quantization, thereby inducing increased quantization errors. In QAT, certain modules fail to learn meaningful representation during the optimization process. This effect and its association with distribution variation have been investigated in AdamBNN (Liu et al., 2021b), where the notion of *flip-flop* was introduced, signifying the change in quantization results of weights at specific iterations. We observed that low-bit transformers are also subject to a comparable effect, termed **oscillation**. This denotes the circumstance where the latent weights fluctuate around the boundary of adjacent quantization bins during quantization-aware training. As per our understanding, Nagel et al. (2022) is one of the few works probing into these effects. However, it restricts its scope to ConvNets and their impact on batch normalization, a technique not employed in transformers. We take the initiative to identify and analyze this oscillation phenomenon specific to transformers.

An illustration of the oscillation phenomenon is shown in Figure 4. Conventionally, the distribution of full-precision initialization adheres to a Gaussian distribution. There exist only a limited number of latent weights that precisely coincide with the optimal quantization value. However, when certain real-value weights $w_t^r$ cross the quantization boundary at iteration $t$, the update of real weights $|w_t^r - w_{t-1}^r|$ triggers an update in the quantized value by a constant value $|q(w_t^r) - q(w_{t-1}^r)| = s$. Here, $s$ represents the quantization scale and constitutes the length of a quantization bin in a uniform quantization scheme. As indicated by the STE detailed in Equation 6, the gradient of the real value is assigned a value identical to this quantized value, resulting in a consistent gradient that encourages the real value to once again traverse the quantization boundary.

We further observe the side effects of transformers in the QAT. As shown in Figure 4a, the weights associated with MHSA tend to accumulate around the quantization threshold following a certain number of epochs. Figure 4b presents an example of this oscillatory behavior within the weights of transformers. This oscillation adversely influences the training and leads to substantial quantization error. The formulation of a solution to prevent this phenomenon through the reduction of variation and mitigation of the impact will be the essential challenge in transformer quantization. In Appendix C, we quantitatively verify our observation across multiple transformer-based models and different layers, proving that weight oscillation phenomenon is a common challenge in transformer QAT.

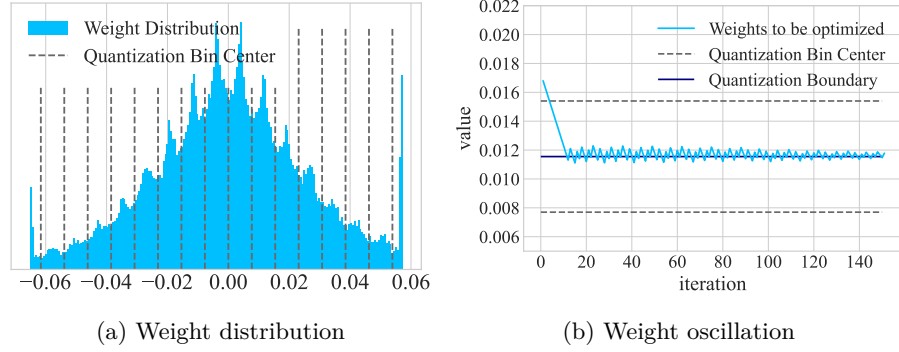

(a) Weight distribution        (b) Weight oscillation

Figure 4: The weight distribution during QAT and the weight oscillation effect due to distribution variance. The layer we select is *blocks.1.attn.proj-v.weight* in 4-bit quantized DeiT-S with scale $\alpha = 0.0077$.

### 4.4 Best Practices of Transformer Quantization

As observed in Section 4, there exists a substantial variation in transformers across three hierarchies: quantization sensitivity, distribution outlier, and weight oscillation. Motivated by these observations, we aim to find the best practices to mitigate the impacts of these variations for effective and efficient low-bit transformer quantization. Our **variation-aware** solutions incorporate several crucial components: a module-dependent quantization scheme, training with variation-aware knowledge distillation, and a regularization strategy to suppress oscillatory behaviors.

#### 4.4.1 Module-dependent Quantization

The scale factor $s$ is the most important parameter in our quantization setting and will be optimized during the quantization-aware training. Our exploration in Section 4.1 establishes a substantial variation in the sensitivity of distinct modules to quantization. However, conventional implementations of transformer quantization often overlook this characteristic. In view of the variability observed in transformers, we propose a module-dependent quantization scheme that facilitates the learning of the quantization scale $s$ at the granular module level (*query*, *key*, and *value* in distinct heads of MHSA). This approach contrasts with previous layer-wise or head-wise quantization methods that assigned a uniform scale to differing modules. Instead, we implement scale-learning quantization at a higher resolution, thereby promoting a finer granularity.

In addition, the outliers in the distribution located in Section 4.2 pose the challenge of an imbalanced gradient scale as these outliers will cause weight updates at a larger scale. Previous work (Bhalgat et al., 2020) has already pointed out the negative impact of an imbalance gradient scale. To overcome this challenge, we adopt a module-wise gradient scaling that balances the weights and scale factor gradient, fully considering the distribution variation in different modules. We multiply the loss of scale factor $s$ by a gradient scale $g$ that encodes the magnitude of the weights in this module, which can be formulated as $\frac{\partial \mathcal{L}}{\partial s} \longleftarrow \frac{\partial \mathcal{L}}{\partial s} \cdot \frac{1}{\sqrt{Q_P ||w||_1}}$, where $||w||_1$ computes the $L_1$-norm of weights in the quantized module. For the modules with higher variation, the $L_1$-norm of weights will be higher than average, and the update of scale factor $s$ will be decreased to ensure that the outliers of the distribution do not influence the scale factor.

Compared to the baseline quantization scheme LSQ, the proposed module-dependent quantization significantly differs from it as we use a **mixed granularity for learnable scale** factors considering the variation in the module sensitivity and **scale the gradient** considering the variation in weight distribution.

#### 4.4.2 Knowledge Distillation

Another practical solution to reduce the variation mentioned in Section 4.2 and help stabilize the training is the Knowledge Distillation (KD) scheme. The core insight of KD is to train our quantized transformer models with a full-precision model as the teacher. The loss function is designed to enforce the similarity between the output distribution of the full-precision teacher and quantized transformer student:

$$\mathcal{L}_{\text{Vanilla}KD} = -\frac{1}{N} \sum_c \sum_{i=1}^N p_c^{T_f}(X_i) \log(p_c^{S_q}(X_i)), \tag{8}$$

where the KD loss is defined as the cross-entropy between the output distributions $p_c$ of a full-precision teacher $T_f$ and a quantized transformer student $S_q$. $X_i$ is the input sample. $c$ and $N$ denote the classes and the number of samples, respectively. Note that one hot label is not involved in training in our setting. The KD scheme helps our model converge fast because it learns the mapping directly from the full-precision teacher, which contains richer information. Previous research (Yuan et al., 2020; Zhou et al., 2020; Menon et al., 2021) also points out that KD loss can be seen as a regularization term to reduce the variance during the training, which makes the training more stable and alleviates the influence of the distribution variation. Here we only employ KD loss as the sole objective to optimize the model, which is more effective with adequate supervision signal in KD (Shen et al., 2021b).

Specifically for ViTs, one disadvantage of the conventional KD scheme is that generating the prediction $p_c^{T_f}$ of the teacher $T_f$ consumes a relatively long time, which makes the training inefficient. To tackle this

Table 3: Comparision with previous quantization methods on BERT-base model and GLUE benchmark. "Bit(W/A/E)" denotes the bitwidth for weights, activations, and embedding.

| Method | Bit(W/A/E) | MNLI(-m/mm) | QQP | QNLI | SST-2 | CoLA | STS-B | MRPC | RTE | Avg |
|---|---|---|---|---|---|---|---|---|---|---|
| FP Baseline | 32-32-32 | 84.9/85.5 | 91.4 | 92.1 | 93.2 | 59.7 | 90.1 | 86.3 | 72.2 | 83.9 |
| BinaryBERT (Bai et al., 2021) | 1-1-1 | 35.6/35.3 | 66.2 | 51.5 | 53.2 | 0 | 6.1 | 68.3 | 52.7 | 53.7 |
| BiBERT (Qin et al., 2021) | 1-1-1 | 66.1/67.5 | 84.8 | 72.6 | 88.7 | 25.4 | 33.6 | 72.5 | 57.4 | 63.2 |
| ReBNN Xu et al. (2023) | 1-1-1 | 69.9/71.3 | 85.2 | 79.2 | 89.3 | 28.8 | 38.7 | 72.6 | 56.9 | 65.8 |
| BEBERT (Tian et al., 2023) | 1-1-1 | 75.7/76.6 | 84.9 | 80.7 | 90.2 | 27.7 | - | 75.1 | 58.6 | 70.5 |
| SGBERT (Ardakani, 2022) | 1-1-1 | 74.3/75.2 | 86.6 | 82.7 | 91.4 | 36.9 | 70.1 | 77.2 | 61.7 | 72.9 |
| BiT (Liu et al., 2022) | 1-1-1 | 79.5/79.4 | 85.4 | 86.4 | 89.9 | 32.9 | 72.0 | 79.9 | 62.1 | 73.5 |
| **Ours** | 1-1-1 | 79.9/79.2 | 87.1 | 86.2 | 91.9 | 36.1 | 73.8 | 80.9 | 59.2 | **74.9 (↑1.4)** |

challenge, we follow Shen & Xing (2021) to utilize a multi-crop KD scheme that first random crops $M$ regions from one image $X_i$, and inputs each cropped image to the teacher model $T_f$ to get the soft label $p_c^{T_f}(X_{i,m}), m \in M$, where $m$ is the index of the cropped region. The soft label is stored together with its coordinates. In the training phase, we directly load the soft label and cropping parameter from the storage for the training with KD. The loss function of this multi-crop KD (MCKD) scheme is:

$$\mathcal{L}_{KD} = -\frac{1}{NM} \sum_c \sum_{i=1}^N \sum_{m=1}^M p_c^{T_f}(X_{i,m}) \log(p_c^{S_q}(X_{i,m})). \tag{9}$$

The higher quality of the soft label generated by this scheme would reduce the variation within a mini-batch to a greater extent. Meanwhile, the data and its corresponding label are loaded the same as the training without knowledge distillation, where the time for inference with the teacher model is saved. We further show in the experiment that this multi-crop KD scheme improves performance by reducing variation and significantly boosts efficiency.

### 4.4.3 Oscillation-aware Bin Regularization

In the analysis of Section 4.3, we identify that the weight distribution variance in the transformer caused oscillation, leading to instability during training. In the view of distribution in each quantization bin, the majority of the weights oscillate between both sides of the quantization bin. To suppress the oscillation during QAT, we regularize the weight distribution with an Oscillation-aware Bin Regularizer (OBR) to encourage the real-value weights to be close to the quantization bin center. The proposed OBR can be formulated as

$$\mathcal{L}_{OBR} = \sum_{m=1}^M (||w_m^r - w_m^q||_2 + \sum_{n=1}^{2^b} \mathcal{V}(w_{n,m}^r)), \tag{10}$$

where $w_m^r, w_m^q, w_{n,m}^r$ represent the real value and quantized value of weights in module $m$, and real value weights in the quantization bin $n$, respectively. $||\cdot||_2$ computes the $L_2$-norm and $\mathcal{V}(\cdot)$ computes variance for all quantization bins with more than two elements. Unlike the previous weight regularization (Chmiel et al., 2020) applied in quantization which only considers the global weight distribution, we minimize the global quantization error and local distribution variance in a specific quantization bin. Ideally, the distribution of the weights in a quantization bin is regularized to be a Dirac delta distribution which can largely suppress the oscillation during training. The final optimization target is $\mathcal{L} = \mathcal{L}_{KD} + \lambda \mathcal{L}_{OBR}$, where $\lambda$ is the weighting coefficient to balance between $\mathcal{L}_{KD}$ and $\mathcal{L}_{OBR}$. To make sure that the regularization does not influence the learning of scale factors at the very early stage of training, we gradually increase the coefficient $\lambda$ during training by applying a cosine annealing schedule following Nagel et al. (2022).

## 5 Experiments

### 5.1 Experimental Settings

**Datasets and Models.** The experiments are carried out on the ImageNet-1K dataset (Deng et al., 2009) and GLUE benchmark (Wang et al., 2018). Our methods are evaluated on DeiT-T (Touvron et al., 2021), SReT-T (Shen et al., 2021a), and Swin-T/S (Liu et al., 2021a) for ImageNet-1K, and BERT-base (Devlin

Table 4: Comparison with previous quantization methods on ImageNet-1K. "Bit-width (W/A)" denotes the bitwidth for weights and activations. "Epochs" denote the total training epochs. The "Baseline" method we use is LSQ+ (Bhalgat et al., 2020).

| Network | QAT Method | Epochs | Apply KD? | Bit-width (W/A) | Top-1 | Bit-width (W/A) | Top-1 | Bit-width (W/A) | Top-1 |
|---|---|---|---|---|---|---|---|---|---|
| DeiT-T (FP Top-1: 73.8) | Q-ViT (Li et al., 2022b) | 300 | ✓ | $4/4^{\dagger}$ | 72.79 | $3/3^{\dagger}$ | 69.62 | - | - |
| | GPUSQ-ViT (Yu et al., 2023) | 300 | ✓ | 4/4 | 71.70 | - | - | - | - |
| | PackQViT (Dong et al., 2023) | 300 | ✓ | 4/4 | 72.70 | - | - | - | - |
| | Baseline | 300 | | 4/4 | 72.62 | 3/3 | 68.22 | 2/2 | 54.45 |
| | Baseline + KD | 300 | ✓ | 4/4 | 73.56 | 3/3 | 69.83 | 2/2 | 56.29 |
| | **Ours** | **150** | ✓ | 4/4 | **74.71 (↑1.15)** | 3/3 | **71.22 (↑1.39)** | 2/2 | **59.73 (↑3.44)** |
| SReT-T (FP Top-1: 75.8) | Baseline | 300 | | 4/4 | 75.65 | 3/3 | 72.59 | 2/2 | 62.11 |
| | Baseline + KD | 300 | ✓ | 4/4 | 76.13 | 3/3 | 74.20 | 2/2 | 64.98 |
| | **Ours** | **150** | ✓ | 4/4 | **76.99 (↑0.86)** | 3/3 | **75.40 (↑1.20)** | 2/2 | **67.53 (↑2.55)** |
| Swin-T (FP Top-1: 81.0) | Q-ViT (Li et al., 2022b) | 300 | ✓ | $4/4^{\dagger}$ | 80.59 | $3/3^{\dagger}$ | 79.45 | - | - |
| | GPUSQ-ViT (Yu et al., 2023) | 300 | ✓ | 4/4 | 80.70 | - | - | - | - |
| | PackQViT (Dong et al., 2023) | 300 | ✓ | 4/4 | 81.50 | - | - | - | - |
| | Li et al. (Li et al., 2022a) | 300 | ✓ | 4/4 | 82.10 | 3/3 | 80.57 | 2/2 | 74.31 |
| | Baseline | 300 | | 4/4 | 80.61 | 3/3 | 79.07 | 2/2 | 70.21 |
| | Baseline + KD | 300 | ✓ | 4/4 | 81.37 | 3/3 | 80.01 | 2/2 | 73.50 |
| | **Ours** | **150** | ✓ | 4/4 | **82.42 (↑0.32)** | 3/3 | **81.37 (↑0.80)** | 2/2 | **77.66 (↑3.35)** |
| Swin-S (FP Top-1: 83.5) | GPUSQ-ViT (Yu et al., 2023) | 300 | ✓ | 4/4 | 83.20 | - | - | - | - |
| | PackQViT (Dong et al., 2023) | 300 | ✓ | 4/4 | 82.80 | - | - | - | - |
| | Baseline | 300 | | 4/4 | 82.45 | 3/3 | 80.57 | 2/2 | 72.38 |
| | Baseline + KD | 300 | ✓ | 4/4 | 82.77 | 3/3 | 81.34 | 2/2 | 73.01 |
| | **Ours** | **150** | ✓ | 4/4 | **83.66 (↑0.46)** | 3/3 | **82.70 (↑1.36)** | 2/2 | **78.11 (↑5.10)** |

$^{\dagger}$ average bitwidth for mixed-precision quantization

et al., 2018) for GLUE. For ViTs, due to the fact that the first (patch embedding) and the last (classification) layer are more sensitive to perturbation compared to intermediate layers, we fix their bitwidth to 8-bit following previous work (Yang & Jin, 2021).

**Training Details.** We adopt real-value pre-trained weights as initialization. The full-precision ViTs are trained from scratch, and the full-precision BERT for various tasks are obtained from public repository[1]. Our baseline quantization method is LSQ+ (Bhalgat et al., 2020). Details of all hyper-parameters and settings are shown in Table 10 and Table 11 in the Appendix.

## 5.2 Comparison with State-of-the-Art Methods

**NLP Tasks.** Table 3 compares our variation-aware solution with existing methods for BERT-base on the GLUE benchmark under binarization (1-bit) setting. The binarized BERT-base with our solution establishes a new state-of-the-art across most tasks in GLUE and improves the average accuracy by 1.4% compared to BiT (Liu et al., 2022).

**Vision Tasks.** Table 4 fairly compares our variation-aware solution with existing methods for DeiT-T, SReT-T, Swin-T, and Swin-S on the ImageNet-1K dataset. We also report the results of baseline LSQ+ with knowledge distillation using the same teacher model to show that performance improvement cannot simply be summarized as learning from the teacher model. Compared with the FP model, our 4-bit quantized DeiT-T achieves 74.71% Top-1 accuracy with a 0.96% absolute gain. Compared with the previous quantization methods, our model also demonstrates remarkable improvement. For example, our 4-bit Swin-T achieves a Top-1 accuracy of 82.42%, which has an absolute gain of 2.83% compared to Q-ViT (Li et al., 2022b). Our method is especially effective for low-precision 2-bit quantization, as our 2-bit Swin-T and Swin-S yields 77.66% and 78.11% Top-1 accuracy, which is 3.35% and 5.1% higher than previous state-of-the-art methods.

**Training Efficiency.** Our methods show better training efficiency on ViTs with the help of a multi-crop knowledge distillation scheme. The better quantization scheme and regularization also help our models converge faster than previous methods with the same training configurations. We only train our models with 150 epochs, sufficient to outperform previous methods with 300 epochs shown in Table 4. The total training time for our DeiT-T with 4 NVIDIA A100 GPUs is 57.3 hours, significantly lower than baseline methods shown in Table 5.

---

[1]https://huggingface.co/textattack/bert-base-uncased-{}

Table 5: Comparison of different teacher models of knowledge distillation for our 4-bit quantized DeiT-T on ImageNet-1K. "Time" indicates the GPU hours of the training process on 4 NVIDIA A100 GPUs.

| Method | Teacher | Top-1 Acc | Top-5 Acc | Time (h) |
|---|---|---|---|---|
| Ours w/o KD | Ground Truth | 72.62 | 91.19 | - |
| Ours w/ Vanilla KD | ResNet152 (He et al., 2016) | 73.56 | 91.52 | 143.5 |
| Ours w/MCKD | ResNet152 (He et al., 2016)
BEiT-L (Bao et al., 2021)
EfficientNet-L2 (Xie et al., 2020) | 74.26
74.49
**74.71** | 91.81
91.92
**92.02** | **57.3** |

### 5.3 Ablation Study

We first perform an overall ablation experiment on 4-bit quantized DeiT-T and 2-bit Swin-T to look into the effectiveness of all proposed modules. The results are shown in Table 6. From the average Standard Deviation of the Absolute Mean (SDAM) and accuracy results, we can see that each module helps alleviate the variation influence and improve the performance of quantized transformers. The following subsections provide a more detailed ablation study of each module.

Table 6: Overall ablation experiment on 4-bit quantized DeiT-T and 2-bit Swin-T. For the experiment "Ours w/o MCKD", the vanilla knowledge distillation with a ResNet152 teacher is applied.

| Method | DeiT-T | | | Swin-T | | |
|---|---|---|---|---|---|---|
| | Top-1 Acc | Top-5 Acc | SDAM | Top-1 Acc | Top-5 Acc | SDAM |
| Ours | 74.71 | 92.02 | 2.13e-2 | 77.66 | 93.68 | 2.97e-2 |
| Ours w/o Multi-crop Knowledge Distillation | 73.56 | 91.52 | 2.30e-2 | 74.15 | 92.20 | 2.95e-2 |
| Ours w/o Module-dependent Quantization | 73.79 | 91.54 | 7.15e-2 | 74.80 | 92.33 | 8.89e-2 |
| Ours w/o Oscillation-aware Bin Regularization | 74.22 | 91.41 | 3.79e-2 | 75.91 | 93.05 | 6.13e-2 |

**Multi-crop Knowledge Distillation for ViTs.** Table 5 compares the Top-1 accuracy of 4-bit quantized DeiT-T without knowledge distillation, with vanilla KD, and with our multi-crop KD of different teachers. The results demonstrate an improvement in both accuracy and efficiency. The teacher model of higher accuracy can improve the performance of student ViTs regardless of architecture. The training time can also be reduced as the soft label is extracted before the training. The time in Table 5 does not include the time for soft label generation, which can be ignored when we have to apply QAT on different models and settings.

**Module-dependent Quantization.** The module-dependent quantization applies a finer-grained quantization scheme at the module level and scales the scale factors' gradients to ensure the scale factor update is not influenced by the variation in transformers. We visualize the loss landscape showing the smoothness of optimization following Li et al. (2018) shown in Figure 6b. Compared to the baseline quantized model, the more centralized and smoother loss landscape reflects that the proposed quantization scheme substantially improves the training stability and efficiency.

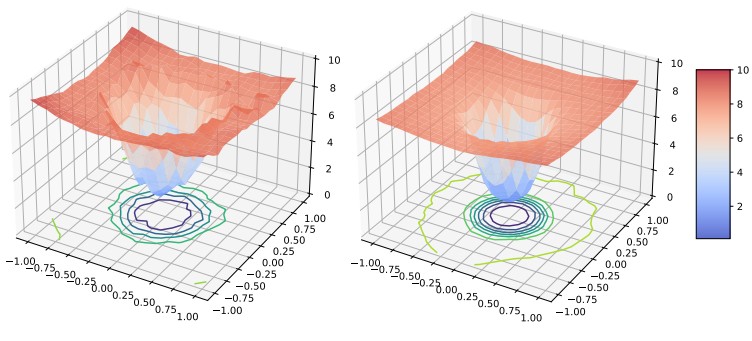

(a) LSQ+ (Bhalgat et al., 2020)    (b) Ours (Module-dependent)

Figure 5: Loss landscape visualization of the 4-bit quantized Swin-T using the baseline (LSQ+ quantization) method and our module-dependent quantization method.

**Oscillation-aware Bin Regularization.** To better know how our oscillation-aware bin regularization can help alleviate the oscillation, we quantify the degree of oscillation during training by measuring the frequency of this phenomenon over time.

We define the oscillation as occurring at iteration $t$ when the quantized integer value changes and the direction of the update in integer value also changes. This can be formulated as:

$$x_t^{\text{int}} \neq x_{t-1}^{\text{int}}, \text{sign}(\Delta_{\text{int}}^t) \neq \text{sign}(\Delta_{\text{int}}^{t^{\text{prev}}}), \tag{11}$$

where $x_t^{\text{int}} = \lfloor \text{clip}(x^r/s, -Q_N, Q_P) \rceil$ is the integer value of input real-value $x^r$ following the notion in Equation 5. The update $\Delta_{\text{int}}^t = x_t^{\text{int}} - x_{t-1}^{\text{int}}$ and $t^{\text{prev}}$ is the iteration of last integer value change. Then the frequency of oscillation is measured using an exponential moving average (EMA):

$$f^t = m \cdot o^t + (1-m) \cdot f^{t-1}, \text{where} \quad o^t = (x_t^{\text{int}} \neq x_{t-1}^{\text{int}}) \wedge (\text{sign}(\Delta_{\text{int}}^t) \neq \text{sign}(\Delta_{\text{int}}^{t^{\text{prev}}})). \tag{12}$$

We define the weights as oscillating weights at iteration $t$ as $f^t > 0.005$. The Top-1 Accuracy of 3-bit quantized SReT-T and the percentage of oscillating weights are shown in Table 7. From the results, we can see a clear negative correlation between weight oscillation percentage and model performance. The proposed Oscillation-aware Bin Regularization (OBR) with a gradually increasing coefficient helps stabilize the training to achieve higher model accuracy.

Table 7: Comparison of 3-bit quantized SReT-T using different regularization. "Oscillation" indicates the percentage of weights that are oscillated at the last iteration of training.

| Regularization | | Top-1 Acc | Top-5 Acc | Oscillation (%) |
|---|---|---|---|---|
| Baseline | | 75.02 | 92.31 | 7.33 |
| KURE (Chmiel et al., 2020) | | 74.85 | 92.24 | 8.12 |
| Ours | $\lambda=\cos(0,1)$ | 75.06 | 92.32 | 0.23 |
| | $\lambda=\cos(0,0.1)$ | **75.40** | **92.49** | 0.78 |
| | $\lambda=\cos(0,0.01)$ | 75.11 | 92.36 | 4.36 |

## 5.4 Hardware Cost Comparison

One direct solution to solve the variation in the quantization sensitivity of different modules is mixed-precision quantization (MPQ), which assigns various bitwidth to different components. However, searching for or learning the optimal bitwidth assignment is difficult and time-consuming, especially for QAT. Among the existing Transformer MPQ methods Liu et al. (2021c); Li et al. (2022b); Xiao et al. (2023b); Ranjan & Savakis (2024); Xu et al. (2024), Q-ViT Li et al. (2022b) is the only work that applies mixed-precision to QAT while the remaining all targets PTQ. Under the same average bitwidth setting, our module-dependent scaling scheme is a more efficient solution to the quantization sensitivity variation than MPQ.

To quantitatively examine the inference efficiency of our method, We compare the hardware utilization of the Q-ViT Li et al. (2022b) MPQ solution with ours, including multiply-accumulate (MAC) units in terms of area and power dissipation. We implemented the MAC operator by Verilog HDL and utilized Cadence Genus to obtain the synthesized area under TSMC 40nm technology and 0.5GHz clock frequency. Specifically, the bitwidth of the partial sum is set to 32bit in case of overflow. The results are listed in Table 8, where the MPQ approach imposes significant overhead in terms of hardware costs compared to our efficient module-dependent scheme. The details of mixed-precision settings are listed in the Appendix.

Table 8: Hardware utilization comparison with MPQ methods for 4/4-bit DeiT-T quantization.

| Quantization Method | Area($\mu m^2$) | Power(mW) |
|---|---|---|
| Ours (Module-dependent Quantization) | 608.404 | 1.589 |
| Q-ViT Li et al. (2022b) (Mixed-precision Quantization) | 893.642 | 1.812 |

## 6    Conclusion

In this work, we have provided a comprehensive understanding of the complexities associated with transformers' low-bit quantization. Through an in-depth analysis of quantization sensitivity, contrasting ConvNets with transformers, and monitoring the weight oscillation during training, we elucidate that the **variation** behavior inherent to transformers poses considerable challenges to low-bit quantization-aware training. To address the challenges presented by variation, we explore the best practice and propose an effective variation-aware quantization technique, including module-dependent quantization and scaling, variation-aware knowledge distillation, and oscillation-aware bin regularization. Through extensive demonstrations, we have shown that our proposed solution to reduce variation in transformers results in state-of-the-art performance across various transformer architectures on both vision and language tasks and significantly improves efficiency.

### Acknowledgments

This research was supported by National Natural Science Foundation of China/HKSAR Research Grants Council Joint Research Scheme under Grant N_HKUST627/20 and by HKSAR RGC General Research Fund (GRF) #16208823.

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

## Appendix

## A    Additional Quantization Sensitivity Analysis

In our paper, the "leave-one-out quantization" experiments are carried out to verify the quantization sensitivity. By quantizing every module except one, a more precise estimation of the real sensitivity of each module to quantization can be achieved. In practical quantization-aware training situations, most modules operate with low precision and are interconnected. Additionally, we provide another form of "quantize-one-module-only" analysis that only quantizes specific parameters. The results are listed in Table 9, which also proves that MHSA is the most sensitive module to the quantization perturbation.

Table 9: Quantize-one-module-only analysis for various components in DeiT-T on ImageNet-1K. The Para(%) stands for the percentage of parameters that are quantized among all trainable parameters.

| Quantization Target | Top-1 Acc(%) | Top-5 Acc(%) | Para(%) |
|---|---|---|---|
| None (FP Model) | 73.75 | 91.87 | 0 |
| All (Baseline 3-bit) | 68.22 | 88.56 | 100 |
| FFN only | 73.51 | 91.72 | 62.1 |
| MHSA only | 72.90 | 91.29 | 31.1 |
| *query* in MHSA only | 73.32 | 91.55 | 7.8 |
| *key* in MHSA only | 73.38 | 91.53 | 7.8 |
| *value* in MHSA only | 73.18 | 91.40 | 7.8 |

## B    Training Details and Dynamics

When training BERT-base across different datasets in the GLUE benchmark, most hyper-parameters are the same. These shared settings include weight decay of 1e-2, warmup proportion of full epochs 10%, linear *lr* scheduler, and Adam optimizer. No data augmentation is applied, and the teacher model for the knowledge distillation is the full-precision pre-trained model of different tasks. For the training epochs, max sequence length, batch size, and learning rate, the settings for different tasks are listed in Table 10.

Table 10: Detailed hyper-parameters and training scheme for different tasks in GLUE benchmark.

| Tasks | MNLI | QQP | QNLI | SST-2 | CoLA | STS-B | MRPC | RTE |
|---|---|---|---|---|---|---|---|---|
| Epoch | 6 | 6 | 20 | 40 | 200 | 40 | 40 | 200 |
| Max Seq Length | 128 | 128 | 128 | 64 | 64 | 128 | 128 | 128 |
| Batch Size | 16 | 32 | 16 | 16 | 64 | 16 | 8 | 32 |
| Initial *lr* | 2e-4 | 2e-4 | 2e-4 | 2e-4 | 5e-4 | 5e-4 | 5e-4 | 5e-4 |

When comparing the results of ViT quantization using our methods with other methods in the experiments, we use the training settings and hyper-parameters shown in Table 11. Generally, most of these hyper-parameters and training settings are the same across different ViT models and different bitwidths settings. We found that applying the proposed Oscillation-aware Bin Regularization (OBR) is more effective for low-bit quantization, including 3-bit and 2-bit. The different performance of OBR among different bitwidth is mainly because penalizing the oscillation during QAT will harm the normal optimization of latent weights, which is more prominent in higher bitwidth. Accordingly, we only apply OBR to the 2-bit and 3-bit quantization.

Fig. 6 compares the training loss and Top-1 test accuracy for 4-bit quantized DeiT-T using our method and LSQ+ (Bhalgat et al., 2020). The core advantages of both effectiveness and efficiency are shown here. In terms of effectiveness, our method can achieve higher Top-1 accuracy and has a more stable loss scheme. For efficiency, our method helps the model converge faster, with only half of the total training epochs.

Table 11: Detailed hyper-parameters and training scheme for different ViT architectures.

| Network | DeiT-T | SReT-T | Swin-T |
|---|---|---|---|
| Epoch | 150 | 150 | 150 |
| Batch Size | 1024 | 640 | 512 |
| Teacher | EfficientNet-L2 (Xie et al., 2020) | EfficientNet-L2 | EfficientNet-L2 |
| Optimizer | AdamW | AdamW | AdamW |
| Initial $lr$ | 5e-4 | 5e-4 | 5e-4 |
| $lr$ scheduler | Consine | Consine | Consine |
| Min $lr$ | 1e-5 | 1e-5 | 5e-6 |
| Warmup $lr$ | 1e-6 | 1e-6 | 1e-6 |
| Weight decay | 1e-4 | 1e-4 | 1e-4 |
| Warmup epochs | 5 | 5 | 5 |
| Random Resize & Crop | ✓ | ✓ | ✓ |
| Random Horizontal Flip | ✓ | ✓ | ✓ |
| Color jittering | - | - | - |
| Number of Crops | 4 | 4 | 4 |

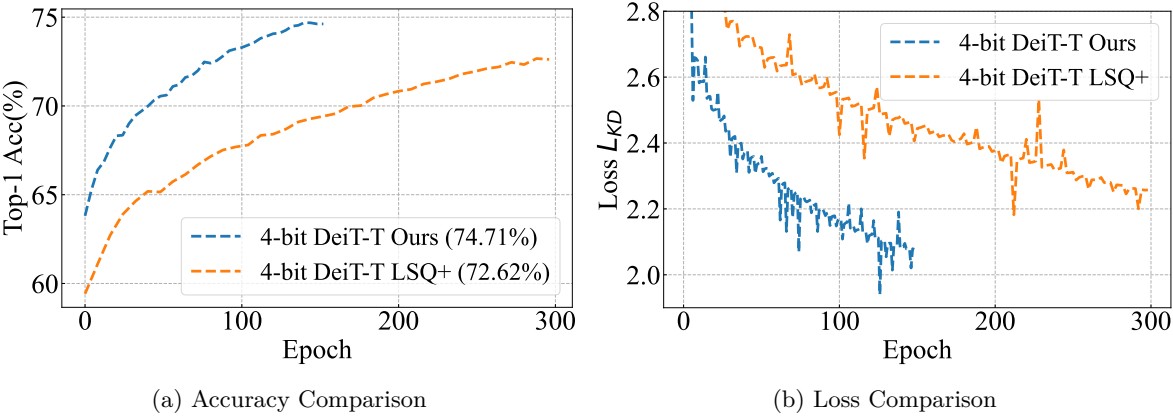

(a) Accuracy Comparison          (b) Loss Comparison

Figure 6: Training dynamics of 4-bit quantized DeiT-T with our methods and LSQ+ (Bhalgat et al., 2020).

## C   Generalizability of the Observations

In Section 4.3, we reveal that the weight and activation distribution outliers will potentially lead to an oscillation phenomenon in QAT, which harms the training stability and quantized model performance. In this section, we further generalize the research target of oscillation phenomenon to various transformer-based models and conduct a finer granularity analysis on the oscillation of different layers. We use the quantitative "oscillation percentage" in QAT defined in Section 4.3 to investigate more transformer-based models and dive into specific layers. The results of the average "oscillation percentage" of DeiT-T (W4A4, W3A3), Swin-T (W2A2), and BERT-base (W1A1E1) are listed in Table 12. We also include ResNet-18 (W4A4) and MobileNetV2 (W2A2) for comparison. The QAT methods are LSQ and all models are initialized from full-precision pretrained models. As can be seen from the results, all transformer-based models suffer from this weight oscillation phenomenon, while it is not observed in ConvNets. These results further prove that weight oscillation phenomenon is a common challenge in transformer QAT.

We also include a per-layer oscillation analysis of Swin-T (W2A2) in Table 13, showing that oscillation phenomenon occurs across all linear weight layers in transformers. Another finding in this per-layer oscillation analysis is that weights in self-attention have a higher oscillation ratio compared to MLP, which corresponds to our conclusion that MHSA is more sensitive to quantization compared to other components.

Table 12: Weight oscillation phenomenon in various transformers and ConvNets.

| Model | Bit Width | Oscillation Percentage (%) |
|---|---|---|
| DeiT-T | W4A4 | 6.69 |
| DeiT-T | W3A3 | 6.97 |
| Swin-T | W2A2 | 5.13 |
| BERT-base | W1A1E1 | 4.31 |
| ResNet-18 | W4A4 | 0.13 |
| MobileNetV2 | W2A2 | 0.20 |

Table 13: Weight oscillation phenomenon in various transformers and ConvNets.

| Layer | mlp.fc1 | mlp.fc2 | attn.qkv | attn.proj |
|---|---|---|---|---|
| Oscillation Percentage (%) | 2.17 | 2.29 | 7.64 | 7.11 |

## D  Attention Map Visualization on ViTs

To demonstrate how our quantization approach preserves the representational capacity of ViT models, we illustrate the attention map of the quantized Swin-T following (Dosovitskiy et al., 2020) and (Abnar & Zuidema, 2020). We fuse the attention heads utilizing maximum operators and exclude low attention pixels to better accentuate the prominent object within the image. As shown in Figure 7, our quantized Swin-T exhibits superior representational capacity by maintaining a more relative ranking within the attention map. This distinction becomes more pronounced when the ViT model is quantized to 3-bit and 2-bit representations. For the baseline LSQ+ quantization (Bhalgat et al., 2020), the attention substantially deteriorates and distributes uniformly across the given input when quantized to extremely low bit-widths. However, our 2-bit quantized Swin-T is still capable of segmenting the salient object region effectively.

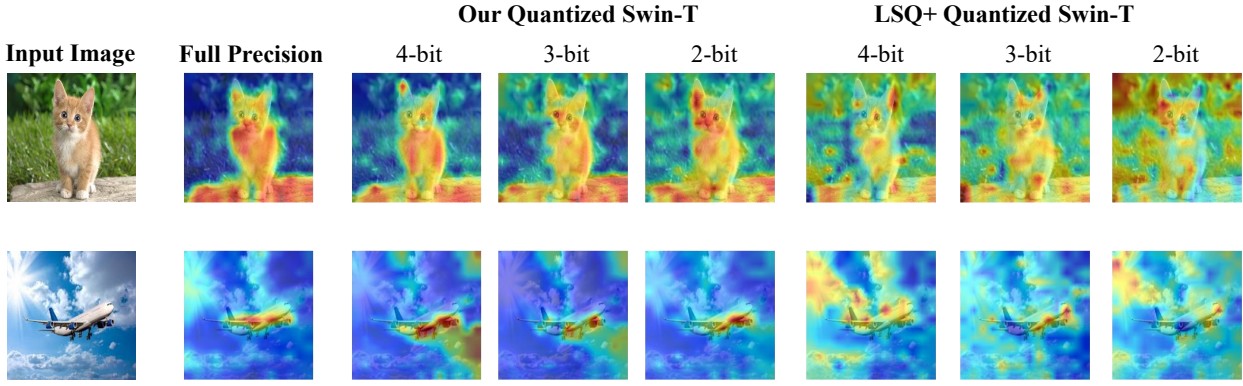

Figure 7: The attention map of quantized Swin-T using our method and LSQ+ (Bhalgat et al., 2020).

## E  Hardware Utilization Experiments Detail

The unit area and power dissipation of multiply-accumulate (MAC) units under different bitwidth settings are listed in Table 14. For mixed-precision quantization schemes such as Q-ViT Li et al. (2022b), the final area should be the maximum area for all bitwidth combinations, and the power dissipation is the weighted average over different settings. Considering the non-linear growth for the power dissipation regarding the bitwidth, a single-precision quantization scheme is better in terms of efficiency than mixed-precision under the same average bitwidth. In addition, most existing mixed-precision accelerators only support power-of-two-bits arithmetic, which poses another challenge for optimal assignment searching or learning.

Table 14: Detailed multiply-accumulate (MAC) unit area and power dissipation of different bitwidth.

| Type | Area($\mu m^2$) | Power(mW) | Type | Area($\mu m^2$) | Power(mW) |
|---|---|---|---|---|---|
| INT2×INT2 | 539.960 | 0.86949 | INT2×INT3 | 551.074 | 0.95939 |
| INT2×INT4 | 562.363 | 1.13939 | INT2×INT5 | 571.360 | 1.30085 |
| INT2×INT6 | 581.062 | 1.41680 | INT2×INT7 | 597.996 | 1.59534 |
| INT2×INT8 | 605.405 | 1.75574 | INT3×INT3 | 571.183 | 1.30043 |
| INT3×INT4 | 589.882 | 1.42975 | INT3×INT5 | 602.053 | 1.57912 |
| INT3×INT6 | 621.634 | 1.69105 | INT3×INT7 | 638.744 | 1.86085 |
| INT3×INT8 | 656.737 | 1.99110 | INT4×INT4 | 608.404 | 1.58901 |
| INT4×INT5 | 635.569 | 1.70870 | INT4×INT6 | 660.089 | 1.85997 |
| INT4×INT7 | 677.200 | 1.94706 | INT4×INT8 | 702.072 | 2.08973 |
| INT5×INT5 | 664.499 | 1.86345 | INT5×INT6 | 695.545 | 2.00091 |
| INT5×INT7 | 718.301 | 2.14442 | INT5×INT8 | 749.347 | 2.24832 |
| INT6×INT6 | 723.593 | 2.12107 | INT6×INT7 | 770.515 | 2.22367 |
| INT6×INT8 | 805.090 | 2.41882 | INT7×INT7 | 817.967 | 2.43294 |
| INT7×INT8 | 864.889 | 2.52819 | INT8×INT8 | 893.642 | 2.67960 |

# F  Overhead of Module-dependent Quantization

Applying quantization-aware training on finer granularity will introduce computation overhead. However, our module-depend quantization (MDQ) only performs per-head-tensor quantization instead of finer granularity, such as per-token or per-channel quantization. This per-head-tensor quantization scheme only introduces minimal trainable parameters (scaling factors) into the training. In addition, the MDQ is only applied to MHSA, and we use layer-wise quantization on feed-forward networks. Table 15 shows the GPU memory consumption and the real training time per epoch on a single NVIDIA 80G A100 GPU with a batch size of 32 on a Swin-T. In addition, the proposed fine-grained quantization scheme helps QAT converge faster. With module-dependent quantization with only 150 training epochs, we can outperform vanilla layer-wise LSQ with 300 epochs. Considering both the performance improvement and faster convergence of QAT, we believe the proposed module-dependent quantization is both efficient and effective.

Table 15: GPU memory consumption and training time per epoch of Swin-T with on a single NVIDIA 80G A100 GPU.

| Quantization Scheme | GPU Memory | Training time |
|---|---|---|
| Layer-wise LSQ | 16.2GB | 40min |
| Module-dependent Quantization | 16.4GB | 43min |

# G  Generalization to ConvNets

One original motivation for this research is to answer the question: *Why can't we directly apply the ConvNet QAT method to Transformers?* This question further leads to other motivations listed in our introduction section. We believe the proposed techniques in this work are novel solutions to the specific variation challenges we located in transformer-based models and will be hard to generalize to ConvNet QAT. More concretely:

- Module-dependent Quantization (MDQ) applies fine-grained quantization by assigning different scale factors to tensors in various heads of MHSA. There is no such module in ConvNets. Therefore, we can not directly apply our MDQ to ConvNet QAT. In addition, previous work on ConvNets Nagel et al. (2021) found that applying channel-wise quantization does not always lead to better performance than layer-wise quantization.

- Multi-crop Knowledge Distillation (MCKD) targets accelerating transformer QAT's convergence. We also agree that this technique will help improve ConvNet QAT. However, MCKD is less useful

for ConvNet mainly because the training of ConvNets is more stabilized, and improving the training stability by reducing the minibatch variation with the better soft label will not significantly improve the performance.

- Oscillation-aware Bin Regularization (OBQ) is specifically designed for transformers based on comparing the weight and activation distribution. The OBQ will not work on ConvNet QAT mainly because weight oscillation and activation outliers are minor challenges in ConvNet architectures.

## H  Additional Discussion with Related Research

**Distribution Outliers** Our analysis of distribution outliers is different from that of SmoothQuant Xiao et al. (2023a). We compare the distribution of transformers to ConvNets and quantify this distribution outlier by the Standard Deviation of the Absolute Mean (SDAM). In contrast, SmoothQuant solely investigates how weights and activations are distributed in LLM and only gives qualitative analysis. In addition, our work targets quantization-aware training (QAT) that could eliminate these outliers by updating weights during gradient-based training. SmoothQuant researches post-training quantization (PTQ), which does not update the model. This further leads to different solutions for our multi-crop knowledge distillation and re-parameterization in SmoothQuant.

**Oscillation** OFQ Liu et al. (2023a) is one research also investigating the training instability of vision transformer. The solutions in Liu et al. (2023a) are the statistical weight quantization (StatsQ) scheme, confidence-guided annealing that freezes the weights, and query-key reparameterization. No solution in this work is similar to our Oscillation-aware Bin Regularization (OBQ), which is simple and effective. In addition, oscillation is only one part of the challenges faced in transformer quantization, and our research provides a more comprehensive analysis, including other variations and performance on transformers for language tasks.

**Quantization Sensitivity and Granularity** While previous research Yang & Jin (2021); Lou et al. (2019) on ConvNet quantization has discussed the granularity of quantization, we would like to highlight that **components** in transformers are very different from ConvNets. While ConvNet basically consists of similar conv and linear layers, transformer-based models comprise more heterogeneous components with diverging characteristics. More concretely, the head-wise quantization in this work is the unique quantization granularity designed for multi-head self-attention modules in transformers, and this head-wise quantization can not be applied to ConvNets. GPTQ Frantar et al. (2023a) is another work exploring the finer-grained quantization for LLMs. While GPTQ Frantar et al. (2023a) employs finer-grain quantization for PTQ, our analysis and solution are designed for QAT. Considering the trainable parameter overhead for the learnable scale and bias in LSQ Bhalgat et al. (2020), the finer quantization granularity used in LLM PTQ, such as per-token in GPTQ and per-channel quantization, are impractical in transformer QAT. Therefore, the head-wise quantization in this research achieves the optimal trade-off between efficiency and performance.

