# OpenReview forum: "Quantization Variation: A New Perspective on Training Transformers with Low-Bit Precision"
_TMLR — Accepted by TMLR_

### Review · Reviewer_ZhyC · 2024-09-01

**Summary Of Contributions:**

The paper proposes a high-precision and high-performance low-bit quantization-aware training scheme tailored for Transformer architectures. Specifically, the authors introduce Module-dependent Quantization, Multi-crop Knowledge Distillation, and Oscillation-aware Bin Regularization.

**Audience:**

Yes

**Broader Impact Concerns:**

None.

**Claims And Evidence:**

Yes

**Requested Changes:**

The authors should check whether the key in MHSA is more sensitive than the value. If the answer is yes, the authors should modify Figure 2(a) correspondingly.

**Strengths And Weaknesses:**

#Strengths

First, through extensive empirical studies, the paper identifies and summarizes the unique variability factors in Transformer architecture quantization, which differ from those in CNN architectures. As a result, the authors design specific optimization mechanisms to address these factors.

Second, extensive experiments were conducted on Transformer-based vision and language models, demonstrating the effectiveness of the proposed optimization mechanisms.

Third, the concerns in the previous round review were satisfactorily addressed:
1) Section 4.4.2 reiterates that the purpose of incorporating Multi-crop Knowledge Distillation is to further reduce variance and enhance efficiency.
2)  Additional experiments were conducted on the Bert-base and ViT-based model architectures.
3) In Table 6, 4-bit quantization experiments for DeiT-T were added to enable comparison with the data in Table 4.

#Weaknesses

The supplementary experiments suggest that the key in MHSA is more sensitive to quantization than the value, which seems to be slightly different from the conclusion in Figure 2(a).

---

> ### Author Response · Authors · 2024-09-09
>
> **W1 & RC1: The authors should check whether the key in MHSA is more sensitive than the value. If the answer is yes, the authors should modify Figure 2(a) correspondingly.**
>
> A: Thanks for your careful investigation and the question! We would like to clarify that the conclusion of the sensitivity analysis is that _value_ are more sensitive to quantization compared to query and key. In the leave-one-out analysis shown in Table 1, the performance are better when _value_ are **not** quantized compared to the case when _key_ or _query_ are **not** quantized. The experiments conducted in Appendix A are quantize-one-module-only analysis, showing that quantizing _value_ in MHSA leads to the most significant accuracy degradation. Both analyses prove that _value_ in MHSA are more sensitive to both _key_ or _query_ to quantization.

---

### Review · Reviewer_tcaJ · 2024-09-03

**Summary Of Contributions:**

The authors investigate how to improve quantization-aware training (QAT) of transformers. They first investigate how quantization of different model components (the feedforward net, MHSA and its subcomponents) differentially impacts performance. They find that the MHSA, and in particular V, are particularly sensitive to quantization, motivating less quantization of these elements. They also find that specific components in specific layers have variable sensitivity. They next investigate an issue where the dynamics of the learning result in the unquantized weights concentrating at the quantization bin borders. They propose a regularization strategy to address this. Finally they consider using knowledge distillation where the quantized model is trained to match a full precision model.

**Audience:**

Yes

**Broader Impact Concerns:**

NA.

**Claims And Evidence:**

Yes

**Requested Changes:**

Editing for clarity and succinctness. I think it could easily be 20% shorter with redundancy removed.

**Strengths And Weaknesses:**

The paper would benefit from editing for clarity and succinctness. I would suggest using Grammarly or Writefull to suggest edits. I found some of the terminology unhelpful, e.g., "variation" can mean so many different things.

The results are not groundbreaking but show a reasonable improvement over existing approaches, even if there is still more to be done to close the gap to the full precision setting. I appreciated the careful ablation analyses in particular.

---

> ### Author Response · Authors · 2024-09-09
>
> **W1 & RC1: Edit the paper for clarity and succinctness. Remove the redundant part and update the unhelpful terminology.**
>
> A: Thanks for the suggestions! We have fully revised the manuscripts to reduce the redundancy and add a more clear explanation of "variation" in the introduction (Section 1) and Method (Section 4). We also fix some typos and grammar issues in the manuscripts.

---

### Review · Reviewer_inoP · 2024-09-04

**Summary Of Contributions:**

The paper addresses the issues posed by unique variation behaviors encountered during low-bit Quantization-Aware Training (QAT) of Transformers, which significantly differ from those observed in ConvNets. Through a quantitative analysis, the study examines the sensitivity of various modules, the presence of outliers in weight and activation distributions, and the oscillation of weight distribution variance during training.

The findings suggest that these variations can induce instability during QAT(Quantization-aware Training) and negatively impact the performance of QAT in Transformer models. To mitigate these issues, the authors propose two novel approaches: module-dependent quantization, which tailors the quantization process based on the sensitivity of individual modules, and oscillation-aware bin regularization, which aims to stabilize weight distribution variance.

The proposed methods are shown to alleviate the oscillations in weight distribution variance during the QAT process, thereby enhancing the performance (accuracy) of low-bit quantized Transformer models across various Transformer architectures.

**Audience:**

Yes

**Claims And Evidence:**

Yes

**Requested Changes:**

- It would be beneficial to include the hyperparameter settings used in the experiments for reproducibility.
- The analyses presented in the paper should be examined to determine whether the observed phenomena are common across general Transformer models or if they vary with different models and precision levels.

**Strengths And Weaknesses:**

[Strengths]
- The paper quantitatively analyzes the sensitivity of Transformer modules and shows that the attention value computation is particularly sensitive to quantization, affecting accuracy.
- The paper proposes module-dependent quantization during Transformer QAT, which allows for different gradient scaling based on the sensitivity of each module.
- Knowledge Distillation (KD) is applied to reduce variation from distribution outliers and stabilize training. The paper also proposes a multi-crop KD method to address training inefficiencies caused by prediction time imbalances between the Teacher and Student (Quantized) models by cropping input activations, showing that this method can enhance training efficiency in KD-based QAT.
- The paper experimentally shows that weight oscillations occur at quantization boundaries during QAT and proposes oscillation-aware bin regularization to reduce oscillations in weight distribution variance during QAT.
- The proposed methods achieve better performance in QAT compared to existing techniques, demonstrating effectiveness in Vision Transformers (ViT), DeiT, and BERT-based models.

[Weaknesses]
- There is a lack of thorough analysis on whether the weight oscillation phenomenon occurring during QAT is specific to certain Transformer models or layers.
- The sensitivity analysis results presented in the paper are conducted for specific Transformer models and bit-precisions. For example, the analysis in Table 1 is performed on 3-bit DeiT. It is unclear whether these differences in module sensitivity occur only in DeiT, only at 3 bits, or if they are a general phenomenon across Transformer models. The experimental results alone are insufficient to determine this.
- The results of the ablation study in Table 6 show that the improvements among the proposed methods are minimal in the 4-bit experiments, making it difficult to discern whether the performance differences are meaningful or just random variations.

---

> ### Author Response · Authors · 2024-09-09
>
> **W1: Whether the weight oscillation phenomenon occurring during QAT is specific to certain Transformer models or layers.**
>
> A: Thanks for this insightful question! We use the quantitative “oscillation percentage” in QAT defined in Section 5.3 to investigate more transformer-based models and dive into specific layers. The results of the average “oscillation percentage” of DeiT-T (W4A4, W3A3), Swin-T (W2A2), and BERT-base (W1A1E1) are shown below. We also include ResNet-18 (W4A4) and MobileNetV2 (W2A2) for comparison. The QAT methods are LSQ and all models are initialized from full-precision pretrained models. As can be seen from the results, all transformer-based models suffer from this weight oscillation phenomenon, while it is not observed in ConvNets. These results are also updated in the manuscript and further prove that weight oscillation phenomenon is a common challenge in transformer QAT.
>
> | Model        | Bit Width | Oscillation Percentage (%) |
> |--------------|-----------|----------------------------|
> | DeiT-T       | W4A4      | 6.69                       |
> | DeiT-T       | W3A3      | 6.97                       |
> | Swin-T       | W2A2      | 5.13                       |
> | BERT-base    | W1A1E1    | 4.31                       |
> | ResNet-18    | W4A4      | 0.13                       |
> | MobileNetV2  | W2A2      | 0.20                       |
>
> We also include a per-layer oscillation analysis of Swin-T (W2A2), showing that oscillation phenomenon occurs across all linear weight layers in transformers. Another finding in this per-layer oscillation analysis is that weights in self-attention have higher oscillation ratio compared to MLP, which corresponds to our conclusion that MHSA is more sensitive to quantization compared to other components.
>
> | Layer        | Oscillation Percentage (%) |
> |--------------|----------------------------|
> | mlp.fc1      | 2.17                       |
> | mlp.fc2      | 2.29                       |
> | attn.qkv     | 7.64                       |
> | attn.proj    | 7.11                       |
>
> **W2: Sensitivity analysis on more transformer models and bit-precisions.**
>
> A: Thanks for your great suggestions. We further conduct the sensitivity analysis on more models and bit width settings, including W2A2 Swin-T and W3A3 SReT-T. The results listed below lead to the same conclusion that multi-head self-attention and _value_ in MHSA are more sensitive to quantization compared to other components.
>
> | Quantization Target           | Swin-T (W2A2) Top-1 Acc | Swin-T (W2A2) Top-5 Acc | SReT-T (W3A3) Top-1 Acc | SReT-T (W3A3) Top-5 Acc |
> |-------------------------------|-------------------------|-------------------------|-------------------------|-------------------------|
> | None (FP Model)               | 81.00                   | 95.25                   | 75.81                   | 91.74                   |
> | All                           | 70.21                   | 85.50                   | 72.59                   | 89.90                   |
> | All, except FFN               | 72.77                   | 88.02                   | 73.06                   | 90.22                   |
> | All, except MHSA              | 77.93                   | 92.59                   | 75.10                   | 91.21                   |
> | All, except _query_ in MHSA   | 73.15                   | 89.74                   | 73.66                   | 90.43                   |
> | All, except _key_ in MHSA     | 73.09                   | 89.98                   | 73.58                   | 90.37                   |
> | All, except _value_ in MHSA   | 75.80                   | 90.66                   | 74.15                   | 90.84                   |

---

> ### Author Response · Authors · 2024-09-09
>
> **W3: The improvements among the proposed methods are minimal in the 4-bit experiments.**
>
> A: Thanks for your great suggestions. We want to classify that the performance of low-bit QAT models are limited by the full-precision performance, especially in higher-precision settings. Under 4-bit experiments, the performance gap between baseline quantized model and full-precision is close (usually under 1%), which leads to a smaller performance improvement brought by our method. We include 2-bit quantized Swin-T ablation experiments here to strengthen that the performance improvements are significant and meaningful.
>
> | Method             | Top-1 Acc | Top-5 Acc | SDAM     |
> |--------------------|-----------|-----------|----------|
> | Ours               | 77.66     | 93.68     | 2.97e-2  |
> | Ours w/o MCKD      | 74.15     | 92.20     | 2.95e-2  |
> | Ours w/o MDQ       | 74.80     | 92.33     | 8.89e-2  |
> | Ours w/o OBR       | 75.91     | 93.05     | 6.13e-2  |
>
> **RC1: It would be beneficial to include the hyperparameter settings used in the experiments for reproducibility.**
>
> A: All the detailed hyper-parameter settings used in the experiments are included in Table 10 and Table 11 in Appendix B. We also add a reference in Section 5.1 in the main text for better clarity.
>
> **RC2: Determine whether the observed phenomena are common across general transformer models and precision levels.**
>
> A: Thanks for the suggestions! In our previous replies, we enhance most analysis and experiments with various models/bitwidth settings to show the generalizability of our conclusions. These new results are included in the main text and a new section in Appendix.

---

### Author Response · Authors · 2024-09-09

We extend our gratitude to all the reviewers for your invaluable comments. We have diligently prepared a thorough response to address all your concerns and updated the manuscripts accordingly to your suggestions.

We are encouraged by the positive comments from reviewers, including empirical and quantitative analysis reveal the sensitivity and unique variability of transformers [Reviewer inoP, ZhyC], a reasonable improvement over existing approaches [Reviewer tcaJ], extensive experiments and careful ablations [Reviewer inoP, tcaJ, ZhyC].

We also thank the reviewers for constructive comments, such as examining whether the observed phenomena are common across general transformers [Reviewer inoP], editing for clarity and succinctness [Reviewer tcaJ], and double-checking the sensitivity of the key in MHSA [Reviewer ZhyC]. We have accommodated all of the comments in our revised manuscripts, where all the revised contents are highlighted in $\textcolor{blue}{blue}$ color.

Detailed responses to each reviewer’s questions, concerns, and requested changes are provided in the following replies. We are eager for you to explore our detailed responses. We always welcome further discussion, and your insights are crucial to refining our paper.

---

### Decision · Action_Editor_yR4m · 2024-09-27

**Recommendation:** Accept as is

**Comment:**

This paper presents an analysis and proposes techniques to improve low-bit Quantization-Aware Training (QAT) of transformer models. The paper first identifies unique challenges compared to CNN models, including module sensitivities, outliers, and oscillation in dynamic weight fluctuations, and then proposes corresponding techniques to address these challenges. The approach is verified across various vision and language transformers, demonstrating improved performance.

The paper received unanimous support from all three reviewers. The reasons for acceptance are:

1). Paper is well-written.

2). It provides a detailed analysis of the variation challenges in transformer models.

3). While the proposed techniques to address quantization challenges are not highly novel, the results showing reasonable improvement over existing approaches.

4). The experiments are extensive.

5). During the rebuttal, additional experimental results and explanations were provided regarding the weight oscillation phenomenon.

Therefore, the paper is recommended acceptance.

**Audience:**

Yes. The paper presents a comprehensive quantitative analysis and extensive empirical results to improve the quantization-aware training of transformer models.

**Claims And Evidence:**

Yes